# Concomitant Administration of Capecitabine and Folate Supplements: Need to Encourage Medication Reconciliation

**DOI:** 10.3390/ph15111388

**Published:** 2022-11-10

**Authors:** Berenice Stefanelli, Carmine Sellitto, Emanuela De Bellis, Martina Torsiello, Nicola Bertini, Angelo Maria Pezzullo, Graziamaria Corbi, Francesco Sabbatino, Stefano Pepe, Angela Tesse, Valeria Conti, Amelia Filippelli

**Affiliations:** 1Postgraduate School of Clinical Pharmacology and Toxicology, University of Salerno, Via S. Allende, 84081 Baronissi, Italy; 2Clinical Pharmacology Unit, San Giovanni di Dio e Ruggi d’Aragona University Hospital, Via San Leonardo 1, 84131 Salerno, Italy; 3Department of Medicine, Surgery and Dentistry “Scuola Medica Salernitana”, University of Salerno, Via S. Allende, 84081 Baronissi, Italy; 4PhD School “Medicina Traslazionale dello Sviluppo e dell’Invecchiamento Attivo”, Department of Medicine, Surgery and Dentistry “Scuola Medica Salernitana”, University of Salerno, Via S. Allende, 84081 Baronissi, Italy; 5PhD School “Scienze Biochimiche e Biotecnologiche”, Department of Precision Medicine, University of Campania “Luigi Vanvitelli”, Piazza Luigi Miraglia, 80138 Naples, Italy; 6Public Health Department, Catholic University, Largo Francesco Vito, 00168 Rome, Italy; 7Department of Medicine and Health Sciences, University of Molise, 86100 Campobasso, Italy; 8Oncology Unit, University Hospital “San Giovanni di Dio e Ruggi d’Aragona”, Via San Leonardo 1, 84131 Salerno, Italy; 9Nantes Université, CNRS, INSERM, l’institut du thorax, F-44000 Nantes, France

**Keywords:** fluoropyrimidines, folic acid, toxicity, adverse drug reactions, hand foot syndrome, diarrhoea

## Abstract

Hand-Foot syndrome (HFS) and diarrhoea are dose-limiting Adverse Drug Reactions (ADRs) of capecitabine-based chemotherapy. Four polymorphisms in the *dihydropyrimidine dehydrogenase* (*DPYD*) gene, encoding the DPD enzyme responsible for the metabolism of fluoropyrimidines, such as capecitabine, are strongly associated with severe ADRs, and their screening should be performed before starting treatment. Moreover, capecitabine-related toxicity may worsen due to drug-drug and drug-supplement interactions. Here we investigated factors responsible for severe HFS and diarrhoea presented by two patients, non-carriers of the recommended *DPYD* single nucleotide polymorphisms (SNPs) but carriers of other genetic variants suggested to increase the risk of capecitabine-related ADRs. Through careful therapy recognition, we demonstrated that, unbeknownst to the oncologists, the patients were taking folic acid during the treatment with capecitabine at a dosage higher than 2000 mg/m^2^, which is the maximum tolerated dose when folate is administered. To resolve the ADRs, the therapy had to be drastically changed. In one case, dose reduction of capecitabine and discontinuation of lipid-lowering agents were carried out. In the other case, discontinuation of capecitabine and folic acid and capecitabine re-administration were performed after a month. Genetic and environmental factors should be considered good predictors of severe capecitabine-related toxicity. Medication reconciliation should be encouraged to avoid the harmful consequences of inappropriate treatments.

## 1. Introduction

Fluoropyrimidines (FPs), such as 5-fluorouracil (5-FU) and capecitabine, are anticancer agents widely used to treat several solid tumours [1]. FPs exert their main anticancer effect by inhibiting thymidylate synthase (TS). The strength of this inhibition is increased by the presence of folates, which work as essential coenzymes for synthesising nucleic acids [2]. Hand Foot Syndrome (HFS) and diarrhoea are common dose-limiting ADRs of FPs. HFS can involve hands and/or feet with symptoms such as numbness, dysaesthesia/paraesthesia, erythema, swelling, blisters, ulceration, desquamation and severe pain [3]. DihydroPyrimidine Dehydrogenase (DPD), encoded by the *DPYD* gene, is the enzyme involved in the catabolism of more than 80% of FPs. More than 160 single nucleotide polymorphisms (SNPs) have been identified in the *DPYD* gene, and it is now recommended to perform pharmacogenetic testing to detect four of them before starting FP therapy [4,5,6,7].

Specifically, the loss-of-function polymorphisms *DPYD c.1905+1G>A* and *DPYD c.1679T>G* almost completely abolish DPD activity, while the variant alleles *c.2846A>T* and *c.1129–5923C>G* are correlated with moderate protein loss function [6,7].

It has been reported that up to 50% of patients carrying *DPYD* SNPs suffer from severe ADRs [8]. However, several patients show FP-related toxicity regardless of the presence of these SNPs, and genetic and nongenetic factors should be considered to prevent even life-threatening ADRs [9,10].

In addition to *DPYD* SNPs, other polymorphisms in other genes involved in the FP pathway or the metabolism of oncologic drugs frequently co-administered with FPs (e.g., oxaliplatin) may be associated with toxicity occurring during an FP-based treatment. The tandem repeat of a 28–base pair (bp) sequence that is present in either duplicate (2R) or triplicate (3R) in the promoter enhancer region of thymidylate synthase (TS) gene (*TSER 2R/3R* polymorphism) has been suggested as a predictive biomarker of capecitabine-related toxicity [11,12]. Other involved polymorphisms implicate genes such as 5,10-methylenetetrahydrofolate reductase (*MTHFR*), cytidine deaminase (*CDA*), Glutathione S-Transferase-P1 (*GSTP1*), Excision Repair Cross-Complementation Group 1 (*ERCC1*) and X-ray Cross-Complementing protein 1 (*XRCC1*) [13,14].

The toxicity associated with chemotherapy, including a capecitabine-based regimen, may worsen due to drug-drug interactions (DDIs) and/or drug-supplement interactions. However, they are often overlooked [15,16]. Indeed, as stated in the Summary of Product Characteristics (SmPC), the maximum tolerated dose (MTD) of capecitabine is 2000 mg/m^2^ per day when used in combination with folinic acid/leucovorin (LV) or folate supplements.

Medication reconciliation is a valuable process for preventing medication errors, including DDI-related adverse events [17]. However, more than 40% of medication errors are believed to result from inadequate reconciliation, especially during the transition of care. Notably, about 20% of these errors are believed to result in harm [18]. Unfortunately, inappropriate use of dietary supplements is not always recognized, especially since these products are often perceived as beneficial substances with no side effects [19,20].

We have described here the steps of a retrospective analysis conducted to investigate factors responsible for the severe toxicity presented by two patients on capecitabine-based therapy. This analysis highlights the need to encourage medication reconciliation in clinical practice.

## 2. Results

### 2.1. Clinical Case of Patient 1

A Caucasian 73-year-old female patient with stage III adenocarcinoma of the right colon had undergone a right hemicolectomy with lymphadenectomy in a University Hospital of Rome. The patient’s medical history included atrial fibrillation and grade II aortic regurgitation. Medications included esomeprazole 20 mg/day, verapamil 40 mg/day, flecainide 100 mg/day, apixaban 2.5 mg/day, ezetimibe 10 mg/day, indapamide 2.5 mg/day and lovastatin 10 mg/day. She was admitted to the University Hospital of Salerno oncology unit to start adjuvant therapy. Based on clinicopathological characteristics and tumour stage, the patient was scheduled for an adjuvant CAPOX chemotherapy regimen consisting of a 2-h intravenous infusion of oxaliplatin (130 mg/m^2^) on day one and oral capecitabine (1000 mg/m^2^, twice daily) from day 1 to 14, every three weeks. Considering her Body Surface Area (BSA), the patient assumed 3000 mg of capecitabine daily and 195 mg of oxaliplatin (Table 1).

After two chemotherapy cycles, the patient reported grade 3 diarrhoea and grade 3 HFS according to the Common Terminology Criteria for Adverse Events (CTCAE) version 5.0 [21]. Therefore, the dose of capecitabine was reduced from 1000 mg/m^2^ twice daily to 625 mg/m^2^ twice daily, and oxaliplatin from 130 mg/m^2^ to 100 mg/m^2^. In addition, the administration of lovastatin and ezetimibe was stopped. Following such changes, the patient’s clinical condition improved and only grade 1 ADRs (nausea, diarrhoea, paraesthesia and conjunctivitis) were reported. After the seventh chemotherapy cycle, according to the RECIST v. 1 [22], a computed tomography (CT) scan demonstrated no tumour lesions, further confirmed by a CT scan performed in September 2020 and March 2021 during follow-up.

By performing a retrospective recognition of all medicines and supplements, the clinical pharmacologists revealed that the oncologists were unaware that the patient was taking iron and multivitamin supplementation before initiating capecitabine-based chemotherapy. Folates were contained in these supplements both as folic acid 400 mcg together with iron 21 mg, vitamin C 75 mg, acerola 300 mg, and as folic acid 100 mcg together with vitamins B1 1.4 mg, vitamin B2 1.6 mg, vitamin B3 18 mg, vitamin B6 3 mg, vitamin B12 1.5 mcg and vitamin B5 9 mg.

Since the patient discontinued the treatment with lipid-lowering agents, the clinical pharmacologists decided to monitor the LDL and total cholesterol levels. Indeed, LDL levels had increased to 107 mg/dL, overcoming the cut-off value of 55 mg/dL, which is recommended for patients at very high risk of cardiovascular disease [23], such as this patient.

The patient was a non-carrier of the recommended *DPYD* SNPs but harboured other polymorphisms suggested to be associated with FP-related toxicity. In particular, she was homozygous for both *TSER 2R/2R* and *MTHFR c.1298CC* genes. The results of all analysed polymorphisms are reported in Table 2.

### 2.2. Clinical Case of Patient 2

A Caucasian 85-year-old female was admitted to the University Hospital of Salerno for mild abdominal pain and nausea. The patient’s past medical history included hypertension and mild cognitive impairment. Abdominal ultrasound and CT scan revealed a right ascending colon tumour lesion. The patient underwent a right hemicolectomy with lymphadenectomy. Histological examination demonstrated Stage II colon cancer. The patient was scheduled for clinical and radiological follow-up based on clinicopathological characteristics. After 18 months of follow-up, a whole-body CT scan demonstrated multiple liver metastasis development. In addition, both levels of CEA and CA 19.9 were increased. Molecular analysis of the primary tumour demonstrated no alterations in *BRAF* or *RAS* genes. Due to advanced age and comorbidities, the patient underwent a chemotherapeutic regimen with reduced doses of capecitabine (800 mg/m^2^) administered on days 1 to 14 every three weeks in monotherapy. Having a BSA of 1.4 m^2^, She assumed 2240 mg of capecitabine daily (Table 1). Further patient’s medication included iron (319 mg/day), enalapril 10 mg/day and aspirin 100 mg/day. Following 5 cycles of capecitabine, a whole-body CT scan demonstrated a partial response according to RECIST v. 1 [22] and normalised serum levels of both CEA and CA 19.9. This response was confirmed on the successive restaging of the disease on capecitabine treatment, and no ADRs were reported. After 15 cycles of capecitabine, according to the CTCAE version 5.0 [21], the patient developed a grade 4 HFS (Figure 1).

Only after the occurrence of the severe ADR, the oncologists revealed that the patient, on her own, had started to assume a multivitamin supplementation, including folic acid 400 mcg, iron 21 mg, vitamin C 75 mg and acerola 300 mg concomitantly with capecitabine.

Treatment with capecitabine and multivitamin supplementation (comprising folic acid) was discontinued, and appropriate supportive treatment was started. In September 2021, a new CT scan demonstrated a further partial response to the therapy with a decrease in the number and diameter of liver tumour lesions. In addition, a clinical examination of the patient reveals the complete resolution of HFS-related symptoms. As a result, capecitabine treatment was restarted, while folate supplementation was avoided. Following four more cycles of capecitabine, no ADRs were reported. A new CT scan was performed, demonstrating a progression of the disease with an increase in the number and diameter of tumour liver lesions. Capecitabine treatment was stopped, and a new treatment with trifluridine/tipiracil was started.

As in the case of patient 1, this patient was a non-carrier of the four *DPYD* SNPs for which screening is recommended but was homozygous for *TSER 2R/2R* gene polymorphism. The results of all analysed polymorphisms are reported in Table 2.

To assess the potential association of all medicines and supplements to the ADRs, a causality assessment was performed using Naranjo Algorithm (NA). Regarding patient 1, a possible association between the reported ADRs and all medicines was found (Table 3).

Regarding patient 2, a probable association between HFS and capecitabine was found both at baseline (with an NA score of 6) and after therapy modifications (with an NA score of 7) (Table 4).

### 2.3. Identification of Drug-Drug and Drug-Supplement Interactions

Five drug interaction checkers (Drugs.com, Drugbank, Medscape, WebMD and Lexicomp) were used to identify potential clinically relevant interactions between folate supplementation and capecitabine.

Among the easy-to-use online tools, four of them are available for free, while one (i.e., Lexicomp) requires payment for consultation and contains “evidence-based” drug-drug interaction (DDI) information [24].

Notably, all of the drug interaction checkers used could have an identified increased risk of toxicity following the concomitant administration of capecitabine and folate supplements.

In particular, Drugs.com and Drugbank revealed a risk for severe DDI-related ADRs when capecitabine is concomitantly administered with folic acid and/or folinic acid/LV. Medscape and WebMD suggested “monitoring closely” potential DDI between capecitabine and LV, while they did not predict interactions with folic acid. Lexicomp provides a distinction based on the severity of the DDI-related outcomes between capecitabine and folic acid (classified as “major”) or capecitabine and folinic acid/LV (classified as “moderate”).

### 2.4. Folates/Capecitabine Use: Alert and Mechanism of Interaction

As already discussed, the SmpC of capecitabine states that toxicity may increase when co-administered with associated folinic acid/LV. In particular, the MTD used in combination with folinic acid/LV is 2000 mg/m^2^ per day. The risk of increased capecitabine-related toxicity may also occur with folate supplements because of the similarity between folinic acid/LV and folic acid [3]. The folinic acid/LV is a naturally reduced form of folate, which is easily converted into the active form, tetrahydrofolate (THF), by the methylenetetrahydrofolate dehydrogenase (MTHFD). In contrast, folic acid is the oxidised synthetic form that requires two enzymatic steps to be converted into the active form by the dihydrofolate reductase (DHFR) [25]. Folate is an essential coenzyme for nucleic acid synthesis. Co-administration with 5-FU and capecitabine determines a pharmacodynamic interaction: FdUMP, one of the 5-FU metabolites, inhibits TS, catalysing the conversion of dUMP in dTMP. FdUMP replaces dUMP and binds to TS in the presence of a reduced form of its natural cofactor (i.e., 5,10-CH2 THF), forming a covalent ternary complex. Notably, the strength of this link is directly proportional to the folate concentration. This pathway is illustrated in Figure 2.

## 3. Discussion

The clinical cases described here suffered from severe diarrhoea and HFS following concomitant administration of capecitabine and folate supplementation. Both patients took a capecitabine dosage greater than 2000 mg/m^2^ per day, which is the MTD of capecitabine when taking concomitant folate administration [3].

Cancer patients often take vitamin supplements, including those containing folic acid, to make up for any deficiencies in the face of increased metabolic requirements imposed by the tumour [26]. DDIs are one of the most important causes of hospitalization but, are often overlooked due to the absence of a complete patient’s pharmacological anamnesis [15]. This problem is highlighted by the World Health Organization and Regulatory Agencies that encourage the performance of medication reconciliation with careful recognition of all treatments to avoid ADRs caused by medication errors, especially during the transition of care [17,27,28].

Both clinical cases highlighted the need to encourage medication reconciliation in clinical practice by stimulating patients to report the use of all drugs and dietary supplements.

Patient 1 had undergone a right hemicolectomy with lymphadenectomy at the University Hospital of Rome and she was already taking folic acid before starting, at the University Hospital of Salerno, the adjuvant CAPOX chemotherapy regimen, while patient two had started to take folic acid independently during capecitabine treatment. In both cases, the use of folate supplements was not recorded.

To resolve the severe diarrhoea manifested by patient one, lovastatin and ezetimibe were discontinued, in addition to a reduced dosage of capecitabine and oxaliplatin. This discontinuation proved detrimental to the patient, who suffered from aortic regurgitation and atrial fibrillation. LDL levels increased, exceeding the reference value for a patient at very high cardiovascular risk (i.e., 55 mg/dL) [29]. This underscores the need for therapeutic appropriateness in patients who follow multitherapy due to multiple comorbidities [30,31,32].

Some studies have suggested that capecitabine toxicity may be increased by previous LV treatment or folate supplements.

Chan et al. conducted a prospective study concerning 144 colorectal cancer patients treated with capecitabine. The authors found grade ≥2 toxicity in nine patients taking folate supplements. The authors found a significant correlation between serum folate levels and toxicity in these subjects, including HFS and diarrhoea [33]. Sharma et al. performed a phase II study in which the main finding was the occurrence of severe toxicity in patients affected by advanced colorectal cancer who had already undergone an adjuvant FU/LV regimen [34].

Yap et al. suggested that serum and red blood cell folate levels predicted capecitabine-induced HFS ≥ grade 2 [35]. In the RCT named PACT (Patient Preference in Adjuvant Colorectal Therapy) by Hennig et al., an increase in treatment toxicity was reported when patients switched from 5-FU/LV regimen to capecitabine. In this study, pre-treatment with 5-FU 425 mg/m^2^ combined with LV 45 mg once weekly triggered toxicity [36]. Among these four studies, three measured serum folates and all identified a mean value of 16 nmol/L as a cut-off associated with worsened capecitabine-related ADRs [33,34,35]. Conversely, the study by Alvarez-Cabellos et al., which enrolled only patients treated with 5-FU co-administered with LV, failed to demonstrate a correlation [26]. In this study, patients had a similar degree of toxicity regardless of serum folate levels. In fact, the safety statement reported in the capecitabine SmPC is not present in that of 5-FU.

Both clinical cases described here were not carriers of the *DPYD* SNPs, for which pharmacogenetic testing is recommended. However, both had the *TSER-2R/2R* genotype.

The polymorphism *TSER 2R/3R* is a 28-base pair of short tandem repeats in the *TS* promoter. As already mentioned, TS catalysed the methylation of dUMP to dTMP, and it is the main target of the FPs [37]. This polymorphism has been associated with capecitabine-related toxicity in patients with advanced colorectal cancer [38,39]. In particular, subjects who are carriers of the *TSER-2R/2R* genotype could be more likely affected by severe toxicity than those who are carriers of the *-3R/3R* genotype. This could be related to a decreased TS mRNA expression in the normal tissue of subjects bearing a *-2R/2R* genotype that may lead to increased TS inhibition by the FP treatment [39,40].

Identifying this polymorphism starting from a peripheral blood sample may be useful in predicting FP-related toxicity, such as HFS and other ADRs, including haematological and gastrointestinal ones [11,41]. Another SNP, referred to as *TS c.58G>C (rs2853542)* in the sequence of the second repeat of the 3R allele, has been associated with FP-related toxicity [42,43]. However, we did not analyse this SNP in the described clinical cases.

The present study also demonstrated the usefulness of interaction checkers in preventing potentially harmful DDI and emphasised the need to use more than one tool. In fact, all five-interaction checkers consulted suggested and identified the potential of the LV/capecitabine combination to worsen capecitabine-related toxicity, but only three of them (i.e., Drugs.com, Drugbank and Lexicomp) extended this warning to folic acid supplementation. This may lead one to underestimate the risk associated with this form of folate, which has a very similar chemical structure to folic acid [44].

The main limitation of the present study is that it is a case report study involving only two patients; thus, the results cannot be generalized. On the other hand, the evaluation of the clinical cases described here underlines the importance of paying special attention to recognising all medications and supplementations taken by patients, especially during transition care.

## 4. Materials and Methods

The clinical cases of two patients were followed in an ongoing observational study (Campania Sud Ethics Committee approval n.4_r.p.s.o./2018) at the University Hospital of Salerno “San Giovanni di Dio e Ruggi d’Aragona” were described. This ongoing study integrated clinical monitoring with a pharmacogenetic analysis conducted in patients eligible for FP-based treatment. ADRs are recorded, and their severity is graded according to the Common Terminology Criteria for Adverse Events (CTC-AE) version 5.0. [21]. The Response evaluation criteria in solid tumours (RECIST) v. 1 is used to assess the treatment response [22]. The enrolled patients gave their informed consent to undergo the procedures in the study, the processing of personal data, and the publication of the results.

The screening of *DPYD* SNPs (*c.1905+1G>A*, *c.1679T>G*, *c.2846A>T* and *c.1129–5923C>G*) is performed at the Clinical Pharmacology Unit of the University Hospital as part of the clinical practice. These SNPs and *CDA c.79A>C* polymorphism were analysed using real-time PCR with allelic discrimination assay (QuantStudio 3, Applied Biosystems™). The genetic variant *TSER 2R/3R* was analysed by PCR with agarose gel-based electrophoresis. The polymorphisms *MTHFR c.677C>T*, *MTHFR c.1298A>C*, *GSTP1 c.313A>G*, *ERCC1 c.*197G>T, ERCC1 c.354T>C* and *XRCC1 c.1196A>G* were identified by pyrosequencing (PyroMark Q96 ID, Qiagen, Hilden, Germany).

All polymorphisms were analysed using commercially available CE-IVD licensed assays (Diatech Pharmacogenetics srl, Jesi, Italy).

A step-by-step retrospective investigation was performed to assess factors possibly responsible for the severe toxicity in the two clinical cases described here.

Naranjo algorithm (NA) was used to evaluate the association between ADRs and patients’ therapy. NA develops an ADR Probability Scale consisting of 10 questions that are answered as either “Yes”, “No”, or “Do not know”. Different point values (−1, 0, +1 or +2) are assigned to each answer. The association drug-ADR is classified as definite, probable, possible, or doubtful. Total scores range from −4 to +13 (definite, score ≥ 9; probable, score 5–8; possible, 1–4; doubtful ≤ 0) [45].

Five drug interaction checkers (Drugs.com, Drugbank, Medscape, WebMD and Lexicomp) were consulted in this study to identify potential clinical ADRs related to drug-drug and drug-supplement interactions.

Written informed consent was obtained from patient 2 for the publication of Figure 1.

## 5. Conclusions

Lack of comprehensive recognition of all drugs and supplements taken by patients can generate harmful consequences. This case report study in which two patients were enrolled allowed us to emphasize that medication reconciliation should be encouraged by considering genetic and environmental factors as valid predictors of serious toxicity. This process is necessary for oncology as in other medical areas and requires the involvement of health care professionals, including clinical pharmacologists.

## Figures and Tables

**Figure 1 pharmaceuticals-15-01388-f001:**
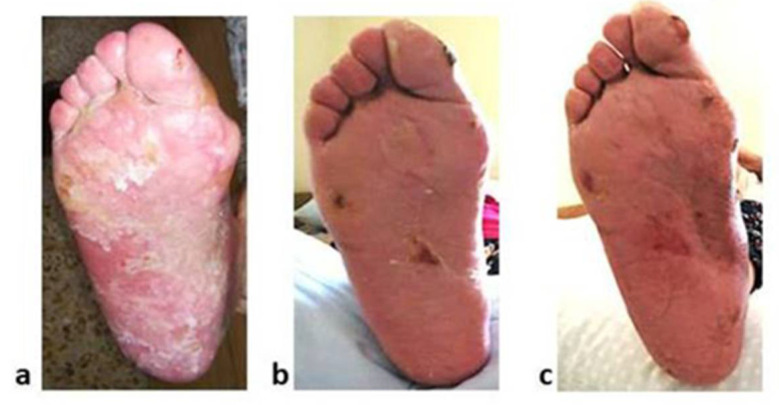
Pictures of grade 4 HFS developed by patient 2. The pictures were taken at different times: (**a**) after 15 cycles of capecitabine when the patient was concomitantly assuming folate supplementation; (**b**) following discontinuation of capecitabine and folate supplementation; (**c**) after capecitabine restarting.

**Figure 2 pharmaceuticals-15-01388-f002:**
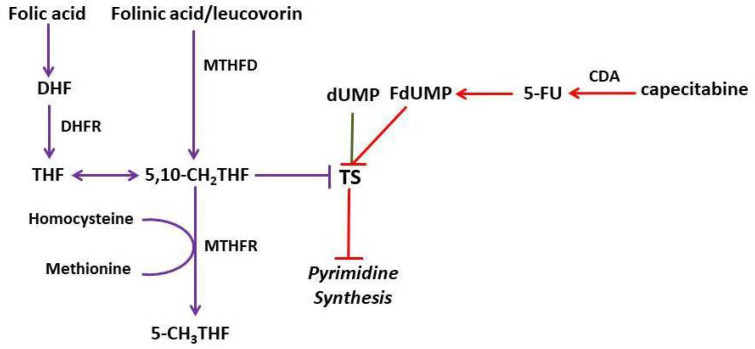
Folate pathway. Abbreviations: DHFR, Diihydrofolate Reductase; THF, Tetrahydrofolate; MTHFD, Methylenetetrahydrofolate dehydrogenase; 5,10-CH2 THF, 5,10-methylenetetrahydrofolate; 5-CH3 THF, 5-Methylenetetrahydrofolate; CDA, Cytidine Deaminase; 5-FU, 5-fluorouracil; dUMP, deoxyuridine monophosphate; TS, Thymidylate synthase; TSER, thymidylate synthase enhancer region; dTMP, deoxythymidine monophosphate (colour figure).

**Table 1 pharmaceuticals-15-01388-t001:** Demographic and clinicopathologic characteristics of patients 1 and 2.

Characteristics	Patient 1	Patient 2
Sex	Female	Female
Age (years)	73	85
BSA (m^2^)	1.5	1.4
Primary tumour site	Colon/rectum	Colon
Tumour stage (AJCC TNM)	III	IV
Chemotherapy regimen	CAPOX (capecitabine 1000 mg/m^2^ BID plus oxaliplatin 130 mg/m^2^)	capecitabine 800 mg/m^2^ BID

Abbreviations: AJCC TNM, American Joint Committee on Cancer Tumour-Node-Metastasis; BSA, Body Surface Area; BID, twice a day.

**Table 2 pharmaceuticals-15-01388-t002:** The table shows the genotypes of patients 1 and 2 based on pharmacogenetic tests performed according to their chemotherapy regimens (CAPOX and capecitabine, respectively). The screening results of the recommended *DPYD*-SNPs (marked with an asterisk) are reported together with those of other polymorphisms proposed as potential predictive factors of FP-related toxicity.

Polymorphisms	Genotype of Patient 1	Genotype of Patient 2
**DPYD c.1905+1G>A* (rs3918290)	GG	GG
**DPYD c.1679T>G* (rs55886062)	TT	TT
**DPYD c.2846A>T* (rs67376798)	AA	AA
**DPYD c.1129-5923C>G* (rs75017182)	CC	CC
*DPYD c.2194G>A* (rs1801160)	GG	GG
*CDA c.79A>C* (rs2072671)	AA	AA
*TSER 2R/3R* (rs45445694)	2R/2R	2R/2R
*MTHFR c.677C>T* (rs1801133)	CC	CC
*MTHFR c.1298A>C* (rs1801131)	CC	AA
*GSTP1**c.313A>G* (rs1659)	AG	-
*ERCC1 c.*197G>T* (rs3212986)	GT	-
*ERCC1 c.354T>C* (rs11615)	TC	-
*XRCC1 c.1196A>G* (rs25487)	GG	-

Abbreviations: *DPYD*, dihydropyrimidine dehydrogenase; *CDA*, cytidine deaminase; *TSER*, thymidylate synthase enhancer region; *MTHFR*, methylene-tetrahydrofolate reductase; *GSTP1*, Glutathione S-Transferase Pi 1; *ERCC1*, excision repair cross-complementing group 1; *XRCC1*, X-ray cross complementing protein 1.

**Table 3 pharmaceuticals-15-01388-t003:** Causality assessment performed using Naranjo Algorithm for patient 1.

Diarrhoea Baseline		No Diarrhoea After Therapy Modification		HFS Baseline		No HFS After Therapy Modification	
DRUG	NA SCORE	DRUG	NA SCORE	DRUG	NA SCORE	DRUG	NA SCORE
capecitabine 1000 mg/m^2^, twice daily	3	capecitabine 625 mg/m^2^, twice daily (reduction)	4	capecitabine 1000 mg/m^2^, twice daily	3	capecitabine 625 mg/m^2^, twice daily (reduction)	4
oxaliplatin 130 mg/m^2^	3	oxaliplatin 100 mg/m^2^ (reduction)	4	oxaliplatin 130 mg/m^2^	3	oxaliplatin 100 mg/m^2^ (reduction)	4
verapamil	3	verapamil	3	verapamil	2	verapamil	2
flecainide	3	flecainide	3	flecainide	2	flecainide	2
apixaban	2	apixaban	2	apixaban	2	apixaban	2
indapamide	2	indapamide	2	indapamide	2	indapamide	2
lovastatin	3	lovastatin (dechallange)	4	lovastatin	2	lovastatin (dechallange)	3
ezetimibe	3	ezetimibe (dechallange)	4	ezetimibe	2	ezetimibe (dechallange)	3
Iron, vitamin C, acerola and folic acid, vitamins B	3	Iron, vitamin C, acerola and folic acid, vitamins B	3	Iron, vitamin C, acerola and folic acid, vitamins B	2	Iron, vitamin C, acerola and folic acid, vitamins B	2

Abbreviations: NA, Naranjo Algorithm; HFS, Hand-Foot-Syndrome.

**Table 4 pharmaceuticals-15-01388-t004:** Causality assessment performed using Naranjo Algorithm for patient 2.

HFS Baseline		No HFS After the First Therapy Modification		No HFS After the Second Therapy Modification	
DRUG	NA SCORE	DRUG	NASCORE	DRUG	NASCORE
capecitabine 800 mg/m^2^, twice daily	6	capecitabine 800 mg/m^2^, twice daily (dechallange)	7	capecitabine 800 mg/m^2^, twice daily (rechallange)	6
folic acid supplement	2	folic acid supplement (dechallange)	3	folic acid supplement (dechallange)	3
Iron supplement	2	Iron supplement	2	Iron supplement	2
aspirin	2	aspirin	2	aspirin	2
enalapril	2	enalapril	2	enalapril	2

Abbreviations: NA, Naranjo Algorithm; HFS, Hand-Foot-Syndrome.

## Data Availability

Data is contained within the article.

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
