# Peer review of "Concomitant Administration of Capecitabine and Folate Supplements: Need to Encourage Medication Reconciliation"

_pharmaceuticals, 2022, doi:10.3390/ph15111388_

Round 1

Reviewer 1 Report (Previous Reviewer 3)

In this revised manuscript, the authors present two cases in which patients were prescribed capecitabine in excess of the tolerable dose when taken in conjunction with folates. Both patients experienced toxicity. In the end, these cases seem like a failure to both properly reconcile all medications and supplements being taken at the time of chemotherapy by medical staff and a failure to properly educate patients about risks associated with supplements taken in conjunction with chemotherapeutics. I do not see how this adds to the body of knowledge in this area since folate interactions with fluoropyrimidines are well documented.

It is appreciated that the authors took the time to respond to comments in the previous review. Many of the critical issues present in the first draft, unfortunately, persist in this version as well. Specifically, the paper is largely unstructured. Materials and methods are not presented at all—there is no methods section. The sections that are present are poorly developed and not written in a way that makes it easy to understand the information that is being presented. It is noted that the authors state that the manuscript was reviewed for language, but many language and grammar issues persist in this draft, which make the text incomprehensible in parts. It still remains unclear what the aim or goal of this work is. The results section mentions that the aim was to assess the association between FP toxicity and SNPs other than 4 commonly tested ones in DPYD (clarified here from the text). This includes genes potentially related to irinotecan and oxaliplatin. How would this association be measured in just 2 patients? What method was used for genotyping (ie, no methods section at all). It is not clear why the genes or (possible) drug:gene interactions in the first paragraph of section 2 (results) were derived. Irinotecan is mentioned. Neither patient received irinotecan. Patient 2 only received capecitabine. So why investigate SNPs related to irinotecan and oxaliplatin as you main aim? This doesn’t match with any of the preceding sections of this manuscript either, including the title. How do the results of this main aim contribute to the conclusions?

 While a signed form allowing the images of HFS is included, nowhere in the text does it at all mention whether informed consent was collected from the patient to participate in this case report or to consent to the use of their clinical information or data for research use. This would have to be overseen by the institution/hospital and would amount to more than an unverified signature on a form. Consent would have needed to be obtained before, not after.

Factual inaccuracies remain in the abstract and elsewhere—there are more than 4 DPYD variants that have been associated with severe ADRs. DPD (not mentioned, but it is the enzyme) does not metabolize capecitabine as stated. It metabolizes a downstream product of capecitabine.

No methods of any kind are included in the manuscript. Sections are 1. Introduction, 2. Case presentation, 3. Discussion, 4. Conclusions.

Multi-component analysis was done according to the abstract. How? Where are the methods? What are you considering to be a multi-component analysis?

Medication reconciliation is standard practice. I’m not sure why the authors are proposing that this is something new and that would need to be done in collaboration with a clinical pharmacologist. The treating physician or pharmacist should be adequate. Failure to properly conduct reconciliation appears to be a main cause of toxicity in both patients because they were treated with too high of a capecitabine dose in conjunction with folate supplementation.

Errors that stem from an incomplete understanding of the literature that were present in the first version persist in this draft. The authors state that there are 4 SNPs associated with FP toxicity. Current guidelines from CPIC and DPWG show that there are far more that have been shown to contribute.

The statistic that 30-50% of patients that receive FPs experience toxicity is without primary support. The “statistic” appears to have been taken from a review article with author overlap with this paper. In that review, that statistic is also not cited. The upper limit of that statistic is not supported by the literature.

The statements regarding TYMS and potential contributions to FP toxicity are outdated and inconsistent with current knowledge in the field.

The introduction does not establish the rationale for this work, nor does establish what the aim of the study was. The summary paragraph/sentence indicates that they are presenting the steps of a retrospective analysis, which differs greatly from the “multi-component analysis” mentioned in the abstract.

The methods for genotyping are not described at all.

The relevance of the variants assessed as “involved in the pathway of FP” is questioned. Please provide support for this statement.

The cases read as though toxicity criteria were graded by the patients instead of the treating physician. Why? How?

The Naranjo Algorithm is not properly explained in the text, nor is it at all described how it was used (see comments about there being no methods section). Therefore it is impossible to assess whether or not it was correctly utilized by the authors. Based on the data presented and what is know, the numbers should not be as compelling as the authors suggest (they remain relatively weak as is). This was an issue with the previous draft and, despite claiming to have provided clarification in the response letter, remains unclear herein.

The pictures of HFS are unnecessary. People who would read this would know what HFS looks like.

Patient 2 did not receive adjuvant therapy. The cancer had progressed and metastasized in the 18 months since surgery. Adjuvant therapy is systemic therapy following curative surgery. Patient 2 received treatment for metastatic disease. There were issues with the understanding of cancer biology and terminology in the previous draft. New ones arose during revision it seems.

The reasons stated in the response and in the text for why curated and peer reviewed approaches for assessing interactions were not used, and online (e.g., webmd) were relied on instead, are insufficient.

The conclusions are not supported by the data. An interaction between TYMS genotype, capecitabine, and folate is indicated, but there is no direct evidence demonstrating an interaction. The conclusion does admit that medication lists were improperly compiled during treatment, which likely is the primary reason behind the toxicity.

A multi-component analysis is highlighted again in the final paragraph of the discussion as being a strength of this paper. A multi-component analysis was not performed.

Author Response

Dear Editor and Reviewer, please find the point-by-point response.

The manuscript was also further revised with regard to the English language.

All changes are highlighted in red.

Reviewer 1

In this revised manuscript, the authors present two cases in which patients were prescribed capecitabine in excess of the tolerable dose when taken in conjunction with folates. Both patients experienced toxicity. In the end, these cases seem like a failure to both properly reconcile all medications and supplements being taken at the time of chemotherapy by medical staff and a failure to properly educate patients about risks associated with supplements taken in conjunction with chemotherapeutics. I do not see how this adds to the body of knowledge in this area since folate interactions with fluoropyrimidines are well documented.

It is appreciated that the authors took the time to respond to comments in the previous review. Many of the critical issues present in the first draft, unfortunately, persist in this version as well. Specifically, the paper is largely unstructured. Materials and methods are not presented at all—there is no methods section. The sections that are present are poorly developed and not written in a way that makes it easy to understand the information that is being presented. It is noted that the authors state that the manuscript was reviewed for language, but many language and grammar issues persist in this draft, which make the text incomprehensible in parts. It still remains unclear what the aim or goal of this work is. The results section mentions that the aim was to assess the association between FP toxicity and SNPs other than 4 commonly tested ones in DPYD (clarified here from the text). This includes genes potentially related to irinotecan and oxaliplatin. How would this association be measured in just 2 patients? What method was used for genotyping (ie, no methods section at all). It is not clear why the genes or (possible) drug:gene interactions in the first paragraph of section 2 (results) were derived. Irinotecan is mentioned. Neither patient received irinotecan. Patient 2 only received capecitabine. So why investigate SNPs related to irinotecan and oxaliplatin as you main aim? This doesn’t match with any of the preceding sections of this manuscript either, including the title. How do the results of this main aim contribute to the conclusions?

Reply: We agree with the reviewer regarding the lack of the "Materials and methods" section. We have added this paragraph as "2. Materials and methods."

This paragraph describes the methods related to the two clinical cases, including genotyping methods. We have mentioned the genetic variants related to the oxaliplatin pathway because they are part of the ongoing study in which the two clinical cases (described in this study) were enrolled. Table 4 shows polymorphisms related to FP and oxaliplatin treatment for patient 1 who received capecitabine and oxaliplatin and only polymorphisms related to FP treatment for patient 2 who received only capecitabine.

  While a signed form allowing the images of HFS is included, nowhere in the text does it at all mention whether informed consent was collected from the patient to participate in this case report or to consent to the use of their clinical information or data for research use. This would have to be overseen by the institution/hospital and would amount to more than an unverified signature on a form. Consent would have needed to be obtained before, not after.

Reply: The patients were enrolled in an ongoing study for which we had already obtained Ethics Committee approval (see section 2. Materials and Methods). In fact, the enrolled patients regularly signed informed consent to publish their data along with those of the other participants.

Then, as requested by the Editorial Board, we also requested and obtained a signed consent form to anonymously publish the image of the HFS. This image is useful in the opinion of the authors.

However, following the reviewer's comment, we also added this statement, " Written informed consent was obtained from patient 2 for the publication of Figure 1." (see Section 2. Materials and Methods).

Factual inaccuracies remain in the abstract and elsewhere—there are more than 4 DPYD variants that have been associated with severe ADRs. DPD (not mentioned, but it is the enzyme) does not metabolize capecitabine as stated. It metabolizes a downstream product of capecitabine.

Reply: We agree with the reviewer that there are more than 4 SNPs associated with FP toxicity. In fact, the purpose of the current study in which the two clinical cases were enrolled is to investigate other polymorphisms possibly related to the FP pathway and the pathway of other drugs co-administered with FP.

The 4 DPYD SNPs (marked with an asterisk in Table 4) are those required before starting FP therapy according to the EMA's 2020 statement (https://www.ema.europa.eu/en/documents/referral/fluorouracil-fluorouracil-related-substances-article-31-referral-annex-iii_en.pdf). To better clarify this, we have added the appropriate references in the introductory section (line 61).

No methods of any kind are included in the manuscript. Sections are 1. Introduction, 2. Case presentation, 3. Discussion, 4. Conclusions.

Reply: We agree with the Reviewer and added the material and methods section as paragraph number 2.

Medication reconciliation is standard practice. I’m not sure why the authors are proposing that this is something new and that would need to be done in collaboration with a clinical pharmacologist. The treating physician or pharmacist should be adequate. Failure to properly conduct reconciliation appears to be a main cause of toxicity in both patients because they were treated with too high of a capecitabine dose in conjunction with folate supplementation.

Reply: Although medication reconciliation should be standard practice, unfortunately, several inappropriate uses of medications and (especially) dietary supplements are not recognized very often. We agree that this important issue was not adequately reported and have added appropriate information and references in the introduction.

In addition, we have changed the title to "Concomitant administration of capecitabine and folate supplements: A need to encourage medication reconciliation."

 Errors that stem from an incomplete understanding of the literature that were present in the first version persist in this draft. The authors state that there are 4 SNPs associated with FP toxicity. Current guidelines from CPIC and DPWG show that there are far more that have been shown to contribute.

Reply: We revised the references and added this sentence, " More than 160 single nucleotide polymorphisms (SNPs) in the DPYD gene have been identified, and it is mandatory today to do pharmacogenetic testing to detect 4 of them before starting FP therapy [4-7]." 

The statistic that 30-50% of patients that receive FPs experience toxicity is without primary support. The “statistic” appears to have been taken from a review article with author overlap with this paper. In that review, that statistic is also not cited. The upper limit of that statistic is not supported by the literature.

Reply: We bag pardon for this mistake. We corrected by specifying that this statistic is referred to patients who are carriers of DPYD polymorphisms.

We changed the sentence to “It has been reported that up to 50% of patients carrying DPYD SNPs suffer from severe ADRs [8]. However, several patients show FP-related toxicity regardless of the presence of these SNPs, and genetic and nongenetic factors should be considered to prevent even life-threatening ADRs [9-10]”.

The statements regarding TYMS and potential contributions to FP toxicity are outdated and inconsistent with current knowledge in the field.

Reply: We disagree with the Reviewer. However, we clarified that this statement needs further investigation (line 341).

The introduction does not establish the rationale for this work, nor does establish what the aim of the study was. The summary paragraph/sentence indicates that they are presenting the steps of a retrospective analysis, which differs greatly from the “multi-component analysis” mentioned in the abstract.

Reply: To further clarify the aim, we introduced some changes in the introduction stressing the importance to encourage medication reconciliation in clinical practice.

The methods for genotyping are not described at all.

Reply: We added this information in paragraph 2 named “Materials and methods”.

The relevance of the variants assessed as “involved in the pathway of FP” is questioned. Please provide support for this statement.

Reply: We added in the manuscript several references providing support for this statement.

The cases read as though toxicity criteria were graded by the patients instead of the treating physician. Why? How?

Reply: We added this information in paragraph 2 named “Materials and methods”.

The Naranjo Algorithm is not properly explained in the text, nor is it at all described how it was used (see comments about there being no methods section). Therefore it is impossible to assess whether or not it was correctly utilized by the authors. Based on the data presented and what is know, the numbers should not be as compelling as the authors suggest (they remain relatively weak as is). This was an issue with the previous draft and, despite claiming to have provided clarification in the response letter, remains unclear herein.

Reply: We added the explanation and reference on the Naranjo algorithm in paragraph 2 named “Materials and methods”.

The pictures of HFS are unnecessary. People who would read this would know what HFS looks like.

Reply: We disagree, as in several case reports the pictures explain very well the adverse event that occurred. We regularly obtained the consent form to show the image.

Patient 2 did not receive adjuvant therapy. Cancer had progressed and metastasized in the 18 months since surgery. Adjuvant therapy is systemic therapy following curative surgery. Patient 2 received treatment for metastatic disease. There were issues with the understanding of cancer biology and terminology in the previous draft. New ones arose during revision it seems.

Reply: Patient 2 had received adjuvant therapy. Subsequently, cancer progressed. In Table 1, we corrected (thanks to the reviewer's advice) that the cancer stage was IV when severe ADR occurred. Also, to avoid confusion, we have removed the treatment intent from Table 1.

The reasons stated in the response and in the text for why curated and peer-reviewed approaches for assessing interactions were not used, and online (e.g., WebMD) were relied on instead, are insufficient.

 Reply: First of all, we specify that the use of these tools was not the only method used in this multi-dimensional analysis.  We know that a peer-reviewed source returns a more realistic result. However, we decided to also use online tools that, unlike peer-reviewed ones, are widely used by clinicians as they are free to use, and with few exceptions, they do not require registration.  All the tools consulted identified the risk of potential toxicity from capecitabine and folic/folinic acid. In addition, we specified that not all tools but only 3/5 (i.e. Drugs.com, Drugbank and Lexicomp) extend this warning to folic acid supplementation. This may lead one to underestimate the risk associated with this form of folate, which actually has a very similar chemical structure to folic acid.

The conclusions are not supported by the data. An interaction between TYMS genotype, capecitabine, and folate is indicated, but there is no direct evidence demonstrating an interaction. The conclusion does admit that medication lists were improperly compiled during treatment, which likely is the primary reason behind the toxicity.

Reply: Following the reviewer's advice and suggestions, we clarified this in the abstract, introduction and conclusion. 

A multi-component analysis is highlighted again in the final paragraph of the discussion as being a strength of this paper. A multi-component analysis was not performed. We agree (it would have been impossible to perform statistical multicomponent analysis with two case reports) and corrected it to “multidimensional evaluation”.

 Multi-component analysis was done according to the abstract. How? Where are the methods? What are you considering to be a multi-component analysis?

Reply: We agree that “multi-component analysis” is a confounding term and changed to “multi-dimensional evaluation”.

Reviewer 2 Report (Previous Reviewer 2)

This study by Stefanelli et.al. is a resubmitted study aiming to describe the steps of a retrospective analysis conducted to investigate factors responsible for the severe toxicity presented by two patients receiving capecitabine-based therapy.

Minor comment:

The number of patients in this study was certainly not sufficient to be regarded as case series (line 325-326). Please revise this.

Author Response

Dear Editor and Reviewers, please find the point-by-point response.

All changes are highlighted in red.

Reviewer#1

This study by Stefanelli et.al. is a resubmitted study aiming to describe the steps of a retrospective analysis conducted to investigate factors responsible for the severe toxicity presented by two patients receiving capecitabine-based therapy.

Minor comment:

The number of patients in this study was certainly not sufficient to be regarded as case series (line 325-326). Please revise this.

Reply: We agree with the Reviewer and changed “case series” to “case report” (now line 321 in the revised manuscript).

Reviewer#2

The authors have improved their manuscript and addressed my previous comments. Their work has merit for publication.

Reply: We thank the Reviewer again for his valuable comments that allowed us to improve our manuscript.

Reviewer 3 Report (Previous Reviewer 1)

The authors have improved their manuscript and addressed my previous comments. Their work has merit for publication.

Author Response

Dear Editor and Reviewers, please find the point-by-point response.

All changes are highlighted in red.

Reviewer#1

This study by Stefanelli et.al. is a resubmitted study aiming to describe the steps of a retrospective analysis conducted to investigate factors responsible for the severe toxicity presented by two patients receiving capecitabine-based therapy.

Minor comment:

The number of patients in this study was certainly not sufficient to be regarded as case series (line 325-326). Please revise this.

Reply: We agree with the Reviewer and changed “case series” to “case report” (now line 321 in the revised manuscript).

Reviewer#2

The authors have improved their manuscript and addressed my previous comments. Their work has merit for publication.

Reply: We thank the Reviewer again for his valuable comments that allowed us to improve our manuscript.

Round 2

Reviewer 1 Report (Previous Reviewer 3)

This is the third version of the manuscript being reviewed. It is appreciated that the authors have added a methods section to this version, brining it more in line with the standard format format for a manuscript of this type. Many of the same issues that were present in the previous two versions persist, however. These include an over-interpretation of the results, mis-representation of the state of regulatory information and information in the literature, and the conclusions not fully fitting with the data. The interpretation of the cases does not appear to have been rigorously performed, despite claims of the authors to the contrary in both the text and the previous rebuttals. For example, this reports two cases, but is now presented as a "multi-dimensional evaluation performed to investigate factors responsible for...". It is not a multi-dimensional analysis (although this is an improvement over the claim about the analytical approach listed in previous drafts. These cases also cannot establish causality or responsibility as alluded to herein. Presenting this as a report of two cases with an accurate interpretation of potential causes (instead of trying to "play up" the methods and results) would make this much more approachable, but after three rounds of revision, I'm not sure that this is possible.

Specific persistent issues:

SNP names are still insufficiently detailed in the manuscript. Variants are presented as a mixture of amino acid changes (without using "p."), nucleotide changes (without using c. or similar appropriate designation), "star" designations, rsIDs, and unspecified names (e.g., 2R/2R). HGVS names should be provided to minimize confusion. See http://varnomen.hgvs.org/ for standards.

The methods used for determining genotype are still inadequately described to assess if they actually evaluated the variants of interest. It is stated that genotyping for some was done in the core as part of clinical testing. How? For others, real-time PCR was used. What primers/probes? For one, gel electrophoresis is stated (using what primers, fragment size, and method of genotype calling). For others, pyrosequencing was used. Again, what primers and how were genotypes discriminated. The information being requested is the minimum that should be presented in a published manuscript.

The methods as presented contain the same issues as in the original draft of the manuscript that was reviewed (prior to deleting the methods for the second draft). Please see and address comments from that review. In short, inadequate information is presented. It is noted that the use of the Naranjo algorithm has been added in this version.

It is noted that written informed consent was obtained for one of the two patients for the publication of a medical image, but it is not indicated that written consent was obtained from patients for publishing these data, or that the study was conducted under the oversight of a regulatory board of any kind.

The picture of HFS is filler and adds nothing to the manuscript, as stated in previous reviews. I understand this is a point of contention with the authors, but as a reader, it is unnecessary. What exactly are you trying to convey with the image? Is it any different that stage 4 HFS that is normally seen in capecitabine use? How does it's inclusion add to the study, beyond that you have seen similar pictures presented previously?

An unacceptable level of grammar issues still persist. It is appreciated that you have again had a native English speaker review this draft. More work is needed on this aspect however. There are many noun–verb inconsistencies. Parentheticals are not applied to the correct subject in sentences throughout, leading to very confusing statements at times. 

The rationale for using non-curated sources of information, e.g., webMD, remains problematic. As stated by the authors in the discussion, all of the information that was needed is part of standard clinical practice and is available in the literature.

The level of association (or causation as implied by the authors herein) that can be attributed to an assessment using the Naranjo algorithm is grossly misrepresented in this manuscript. This is not analytical tool, as suggested herein, and should not be presented or used as such, especially when clear cut information is available.

There are multiple inconsistencies with the use of terms. For example, patients were taking multivitamins containing folic acid. These are also called "natural products" in various places in the manuscript, and the conclusions are specifically drawn to natural products. While I agree that supplements, vitamins, and natural products should be part of medication reconciliation, these are not necessary equivalent and cannot be used interchangeably.

The new statement about "the need for therapeutic appropriateness" is unclear. What is meant by this term? 

"Worsening" suggests a progression of toxicity. That does not appear to be the case here.

The description of how these observations fit into the literature is still superficial. Previous studies are described, but how this adds to the previously published and already known body of literature is unclear.

The relevance of data from red blood cell counts in the Yap study to the current study is unclear.

Did the Hennig study actually study supplement use or just folate levels? Again, as written, how this study supports or contradicts those findings is unclear, since I do not believe actual folate levels were measured in this study.

Again, Naranjo does not demonstrate causality as stated on table 3. Regardless, the (subjective) scores would be weakly suggestive of an interaction based on the data shown. It's not clear what presenting this above just a logical assessment adds to the manuscript.

Despite arguments to the contrary in the previous rebuttal letter, the "2R" nomenclature for TYMS TSER is outdated. The authors are again encouraged to read up on current literature about the TSER and what is actually contributing to expression of TS and toxicity risk in the TSER (it is not just 2R/3R/etc repeat status).

It would seem that the harmful changes to the treatment regimen of patient 1 were independent of the interaction and due to a choice made by the treatment team. It is not clear how this information adds to the body of knowledge as written in the discussion.

The rationale behind the authors assumption that mode of administration for FPs is related to the interaction with folates and toxicity. The authors are encouraged to study the compounds contributing to HFS in capecitabine and why those same compounds would not be a concern in 5-FU. As written, this makes very little sense form a mechanistic point of view when the actual link seems pretty clear.

Line 328: another grammar error. "carriers of the DPYD SNP." which SNP?

line 328-329. Testing for DPYD SNP is not mandatory. This seems to be a persistent mis-interpretation of the EMA statement on this matter. The authors are encouraged to present information that is consistent with the statement that is actually published by the EMA. If different (localized) guidelines are also in play, that fact is not made clear anywhere herein.

The statement about invasiveness of testing for TS variants in lines 339-342 is very confusing. Why is testing for a germline variant considered invasive? The earlier work is in regard to toxicity, which presumably considered germline since "normal" tissue is mentioned. Looking at the cited papers, it does not seem the authors have correctly interpreted those results. Please revise accordingly and/or clarify what is actually meant here.

The statement about needing large studies for TSER interpretation is inconsistent with what is known in the literature, as pointed out in the previous reviews. Please reference those reviews and update this section with current information.

Line 345-251: The issue does not appear to be related to the use of interaction checkers as stated here. Instead it seems to be due to obtaining an incomplete medication report and inadequate patient education.

Same lines: the risk was due to folic acid supplementation. It seems to reason that the structure of folic acid is similar to that of folic acid (as stated here) since they are the same compound. Again, what does this add?

What variant was actually studied for the TSER. rs183205964 is a SNP the 5; UTR of TYMS, but as far as I can tell from the rest of the manuscript, that is not the variant that was tested. Figure 2 contains this information.

The directionality of information on Fig 2 is confusion. Are you saying that TS blocks FdUMP to dUMP, which doesn't actually occur? Are you trying to convey that rs183205964 blocks DNA synthesis? If TS function is blocked, dTMP pools become depleted, but TS does not directly participate in DNA synthesis (ie, replication and repair) as indicated. dTMP would though.

Author Response

This is the third revised version of the manuscript (in addition to the revision completed by the other two reviewers who gave approval for publication).

However, wanting to take the opportunity to further improve our manuscript, we have carefully drafted this third round of revision and have tried to give another adequate point-by-point response, with the changes highlighted in red in the manuscript.

Reviewer 1          

This is the third version of the manuscript being reviewed. It is appreciated that the authors have added a methods section to this version, brining it more in line with the standard format format for a manuscript of this type. Many of the same issues that were present in the previous two versions persist, however. These include an over-interpretation of the results, mis-representation of the state of regulatory information and information in the literature, and the conclusions not fully fitting with the data. The interpretation of the cases does not appear to have been rigorously performed, despite claims of the authors to the contrary in both the text and the previous rebuttals. For example, this reports two cases, but is now presented as a "multi-dimensional evaluation performed to investigate factors responsible for...". It is not a multi-dimensional analysis (although this is an improvement over the claim about the analytical approach listed in previous drafts. These cases also cannot establish causality or responsibility as alluded to herein. Presenting this as a report of two cases with an accurate interpretation of potential causes (instead of trying to "play up" the methods and results) would make this much more approachable, but after three rounds of revision, I'm not sure that this is possible.

Reply: We have tried to improve our manuscript without compromising the important message we intended to give. The term multidimensional has been removed along with several other changes that are highlighted in red throughout the manuscript.

Specific persistent issues:

SNP names are still insufficiently detailed in the manuscript. Variants are presented as a mixture of amino acid changes (without using "p."), nucleotide changes (without using c. or similar appropriate designation), "star" designations, rsIDs, and unspecified names (e.g., 2R/2R). HGVS names should be provided to minimize confusion. See http://varnomen.hgvs.org/ for standards.

Reply:  We used the suggested nomenclature to indicate the SNPs. Regarding the VNTR TSER 2R>3R, we specified as follows: “The tandem repeat of a 28–base pair (bp) sequence that is present in either duplicate (2R) or in triplicate (3R) in the promoter enhancer region of thymidylate synthase (TS) gene (TSER 2R/3R polymorphism) (See lines 72-74).

The methods used for determining genotype are still inadequately described to assess if they actually evaluated the variants of interest. It is stated that genotyping for some was done in the core as part of clinical testing. How? For others, real-time PCR was used. What primers/probes? For one, gel electrophoresis is stated (using what primers, fragment size, and method of genotype calling). For others, pyrosequencing was used. Again, what primers and how were genotypes discriminated. The information being requested is the minimum that should be presented in a published manuscript.

Reply: For pharmacogenetic analyses performed at our Clinical Unit, we use commercially available CE-IVD-licensed real-time PCR and pyrosequencing assays. This information has been added in the methods section.

The methods as presented contain the same issues as in the original draft of the manuscript that was reviewed (prior to deleting the methods for the second draft). Please see and address comments from that review. In short, inadequate information is presented. It is noted that the use of the Naranjo algorithm has been added in this version.

It is noted that written informed consent was obtained for one of the two patients for the publication of a medical image, but it is not indicated that written consent was obtained from patients for publishing these data, or that the study was conducted under the oversight of a regulatory board of any kind.

Reply: According to the Reviewer’s suggestion, we have made this information explicit. The two patients were enrolled in an ongoing study, already described in the "Methods" section. Their enrollment in the study required informed consent, by which the patient consents to the procedures involved in the study, to the processing of personal data, and to the publication of the results. The study was approved by the Campania Sud Ethics Committee and identified with code no.4_r.p.s.o./2018 (See lines 100-108).

The picture of HFS is filler and adds nothing to the manuscript, as stated in previous reviews. I understand this is a point of contention with the authors, but as a reader, it is unnecessary. What exactly are you trying to convey with the image? Is it any different that stage 4 HFS that is normally seen in capecitabine use? How does it's inclusion add to the study, beyond that you have seen similar pictures presented previously?

Reply:  We can understand that, according to the reviewer (#1), the picture of the HFS is filler and adds nothing to the manuscript. However, in our opinion and that of the other two reviewers who have already approved the manuscript for publication, the picture supports the detailed narrative of our clinical case (as is very often the case in other case reports). The proposed picture represents a very important piece of data for understanding the patient's clinical condition, which improves markedly after changes in therapy. Only viewing the foot through the picture can account for the severity of ADR and its subsequent improvement, without offending anyone!

An unacceptable level of grammar issues still persist. It is appreciated that you have again had a native English speaker review this draft. More work is needed on this aspect however. There are many noun–verb inconsistencies. Parentheticals are not applied to the correct subject in sentences throughout, leading to very confusing statements at times. 

Reply: The reviewer is thanked for the advice. Further revision of the English language has been made.

The rationale for using non-curated sources of information, e.g., webMD, remains problematic. As stated by the authors in the discussion, all of the information that was needed is part of standard clinical practice and is available in the literature.

Reply: We believe that the consultation of DDI checkers is part of clinical practice. Therefore, it is useful to highlight the pros and cons of using such tools. In the discussion section, we highlighted the usefulness of interaction checkers in preventing potentially harmful adverse events related to DDIs and, at the same time, the need to use more than one tool.

The level of association (or causation as implied by the authors herein) that can be attributed to an assessment using the Naranjo algorithm is grossly misrepresented in this manuscript. This is not analytical tool, as suggested herein, and should not be presented or used as such, especially when clear cut information is available.

Reply: NA was used to test the association between medications taken by patients and ADRs as is commonly done in the pharmacovigilance system. However, we agree that the text can generate misunderstanding and decided to move the narration (and tables order) regarding NA after the description of the clinical cases (See lines 234-240). Moreover, we eliminated the comment on NA from the discussion.

There are multiple inconsistencies with the use of terms. For example, patients were taking multivitamins containing folic acid. These are also called "natural products" in various places in the manuscript, and the conclusions are specifically drawn to natural products. While I agree that supplements, vitamins, and natural products should be part of medication reconciliation, these are not necessary equivalent and cannot be used interchangeably.

Reply: In the previous version of the manuscript, “Natural products” was mentioned only once. However, we changed this to “dietary supplements” (lines 299-300).

The new statement about "the need for therapeutic appropriateness" is unclear. What is meant by this term? 

Reply: This statement has been better clarified (see lines 306-312).

"Worsening" suggests a progression of toxicity. That does not appear to be the case here.

Reply:  "Worsening” has been changed to “may worsen” (see lines 32-33, abstract and line 80, introduction).

The description of how these observations fit into the literature is still superficial. Previous studies are described, but how this adds to the previously published and already known body of literature is unclear.

The relevance of data from red blood cell counts in the Yap study to the current study is unclear.

Did the Hennig study actually study supplement use or just folate levels? Again, as written, how this study supports or contradicts those findings is unclear, since I do not believe actual folate levels were measured in this study.

Reply: As discussed, four studies suggested a possible link between the increase of capecitabine toxicity and folate levels. However, only three of them also measured serum or red blood cell folate levels.

All studies suggest that the cause of capecitabine-related increased toxicity could be due to previous LV intake or folate supplement use.

To better clarify this issue, we added the following sentence: “Some studies have suggested that capecitabine toxicity may be increased by previous treatment with LV or the use of folate supplements” (lines 313-314).

Again, Naranjo does not demonstrate causality as stated on table 3. Regardless, the (subjective) scores would be weakly suggestive of an interaction based on the data shown. It's not clear what presenting this above just a logical assessment adds to the manuscript.

Reply: Please, see the previous reply regarding this point.

Despite arguments to the contrary in the previous rebuttal letter, the "2R" nomenclature for TYMS TSER is outdated. The authors are again encouraged to read up on current literature about the TSER and what is actually contributing to expression of TS and toxicity risk in the TSER (it is not just 2R/3R/etc repeat status).

Reply: We used the nomenclature referred to as “repeat number” corresponding to rs45445694. Similarly, other studies highlight a significant association of 2R allele carriage (2R/2R or 3R/2R+2R/2R genotypes) with FP-related toxicity of grade 3-4. We have already reported two of these studies (ref 12 and 43).

However, we agree on the need to analyse also the G > C SNP (rs2853542) that indeed we are planning to analyse in another study in a larger study population. At the same time, we would like to underline that the tandem repeat of 28 bp TSER 2R/3R (rs45445694) and the SNP TS c.58 G>C (rs2853542) both lead to reduced TS expression (as stated in the studies corresponding to the references 43 and 44). Thanks to the Reviewer’s persistent advice, we have had the opportunity to add also the information on the SNP TS c.58 G>C in our manuscript (lines 346-349).

It would seem that the harmful changes to the treatment regimen of patient 1 were independent of the interaction and due to a choice made by the treatment team. It is not clear how this information adds to the body of knowledge as written in the discussion.

Reply: We agree with the Reviewer. To avoid possible misunderstanding we changed the sentences (in the previous version of the manuscript in lines 297-300) to: “To resolve the severe diarrhoea manifested by patient 1, lovastatin and ezetimibe were also discontinued in addition to reduced dosage of capecitabine and oxaliplatin. This discontinuation proved to be detrimental to the patient who suffered from aortic regurgitation and atrial fibrillation.” (lines 306-309).

The rationale behind the authors assumption that mode of administration for FPs is related to the interaction with folates and toxicity. The authors are encouraged to study the compounds contributing to HFS in capecitabine and why those same compounds would not be a concern in 5-FU. As written, this makes very little sense form a mechanistic point of view when the actual link seems pretty clear.

Reply: We thank the Reviewer for his/her kind invitation to study! As suggested, we studied this topic and verified that there is no conclusion on possible mechanisms. However, to not ignore the reviewer's advice, we eliminated our hypothesis (previously expressed as "could depend on…").

Line 328: another grammar error. "carriers of the DPYD SNP." which SNP?

Reply: Sorry for such an error, we have corrected this.

line 328-329. Testing for DPYD SNP is not mandatory. This seems to be a persistent mis-interpretation of the EMA statement on this matter. The authors are encouraged to present information that is consistent with the statement that is actually published by the EMA. If different (localized) guidelines are also in play, that fact is not made clear anywhere herein.

Reply: We changed “mandatory” to “recommended”.

The statement about invasiveness of testing for TS variants in lines 339-342 is very confusing. Why is testing for a germline variant considered invasive? The earlier work is in regard to toxicity, which presumably considered germline since "normal" tissue is mentioned. Looking at the cited papers, it does not seem the authors have correctly interpreted those results. Please revise accordingly and/or clarify what is actually meant here.

Reply: We apologize for using the word “germline” incorrectly. So we have removed that word from the sentence (lines 344-345) .

The statement about needing large studies for TSER interpretation is inconsistent with what is known in the literature, as pointed out in the previous reviews. Please reference those reviews and update this section with current information.

 Reply: We changed the sentence as follows: Another SNP, referred to as TS c.58G>C (rs2853542) in the sequence of the second repeat of the 3R allele has been associated with FP-related toxicity [43,44]. However, we did not analyse this SNP in the described clinical cases (lines 346-349).

Line 345-251: The issue does not appear to be related to the use of interaction checkers as stated here. Instead it seems to be due to obtaining an incomplete medication report and inadequate patient education

Same lines: the risk was due to folic acid supplementation. It seems to reason that the structure of folic acid is similar to that of folic acid (as stated here) since they are the same compound. Again, what does this add?

Reply: We have discussed extensively that the main cause is incomplete drug reports. However, we prefer to remain free to suggest the existence of other variables possibly associated with ADRs. The risk alert, included in capecitabine SMPc, is extended also to folinic acid. This is an important issue. In addition, it is crucial to remark, as discussed by these authors, that the use of more than one interaction checker (tools used in clinical practice) is needed to identify potentially harmful DDI. 

What variant was actually studied for the TSER. rs183205964 is a SNP the 5; UTR of TYMS, but as far as I can tell from the rest of the manuscript, that is not the variant that was tested. Figure 2 contains this information

The directionality of information on Fig 2 is confusion. Are you saying that TS blocks FdUMP to dUMP, which doesn't actually occur? Are you trying to convey that rs183205964 blocks DNA synthesis? If TS function is blocked, dTMP pools become depleted, but TS does not directly participate in DNA synthesis (ie, replication and repair) as indicated. dTMP would though.

Reply: There was an error in Figure 2. Actually, the genetic variant analyzed in this study was SNP rs45445694. However, agreeing that Figure 2 needed to be improved, we modified it to reflect the mechanism of action of capecitabine by also removing the reference to genetic variants (see Figure 2).

This manuscript is a resubmission of an earlier submission. The following is a list of the peer review reports and author responses from that submission.

Round 1

Reviewer 1 Report

In the present work the authors have intended to conduct an integrated clinical pharmacological analysis to shed light on genetic and environmental predictors of capecitabine-related severe toxicity. They have also attempted to perform a systematic review in order to compare their observations to the reported adverse effects in the literature.

The topic is interesting, yet there are several issues that does not make it suitable for publication in its present form.

The article is a bit confusing. On one hand the authors presented two patient cases of Capecitabine adverse effects and on the other hand they present a systematic review. It appeared that the contents were merged in order to increase the manuscript in size, and  not because it was necessary for presenting their case. It would be more useful to the medical community if they would separate the clinical cases and present them as case studies, and then conduct a systematic study on their topic. I really do not see the point of presenting observational results (such as the clinical cases, but also experimental procedures) along with a literature study.

This is probably the reason, why I found the aim of the study confusing at the very beginning of the text, where the aim is not clear and not well-presented on what they intend to do.

For those reasons, their manuscript is not suitable for publication in its present form. The authors should separate the clinical observations from the review and provide two manuscripts for this topic.

Reviewer 2 Report

The study aimed to determine genetic and non-genetic factors that contribute to capecitabine-related severe toxicity, including co-administration of folic supplements.

It is a study with an interesting topic as the impact of drug-drug/supplement interactions in drugs with actionable pharmacogenetics (i.e. 5-FU) variants is often left unnoticed.

Minor comments:

·       - Line 132: “./die”. Do you mean “day”? Please recheck the manuscript for typos.

·       Table 3 (Patient 2) is badly placed. It is not clear to the reader.

·       Repetition in “Lipidic profile” in Table 4.

·       Line 295-299: Why do the authors report only the 95% CI of effect size? What are the main effect sizes? Please report the meta-analyses result in a standard form (including the I2).

·       Please report the meta-analyses result in a standard forest plot.

·       Please discuss the strength and limitations of the study, especially since this is a case series.

·       Line 442-446: Please mention the what are variables that contributed to capecitabine-related ADRs based on this study.

Major comments: Why was there no risk of bias analysis in the systematic review?

Reviewer 3 Report

In the reviewed manuscript, Stefanelli et al., present two cases of hand-foot syndrome and diarrhea that occurred in patients undergoing cancer therapy with capecitabine-based regimens. These toxicities are common in capecitabine use. The two cases were also taking multivitamin supplements that contained folate during their treatment. Folate supplementation is contraindicated in capecitabine use because of well-documented associations with increased risk of adverse drug reactions to capecitabine. Folinic acid (leucovorin), is often co-administered with 5-FU (the compound formed by activation of capecitabine). Leucovorin is similarly converted to reduced folate derivatives (methylentetrahydrofolate) that act as cofactors for 5-FU metabolites to bind TS forming an inactive ternary complex. Folate and folinic acid are known to potentiate the anti-cancer effects of 5-FU metabolites, as well as increasing off-target toxicities as a consequence. Folinic acid is typically used clinically because fewer steps are needed to reach methyletetrahydrofolate.

The major limitation of the manuscript is that it is unclear what this adds to the field. The case reports do not draw a clear line between folate supplementation and the toxicity. The authors performed genetic tests, which also do not clearly indicate any type of association. The authors also performed a "causality assessment" that is usually used to try to find direction when no information is known that links symptoms to treatment. In this case, the added value of the assessment is unclear as it indicates toxicity is likely due to capecitabine, which is known; however, the authors seem to conclude from that assessment that the link is to supplemental folate, despite the data showing capecitabine. Lipid profiles are shown for 1 patient because the treating physician discontinued statin therapy. This aspect is not really relevant to the topic of this paper, yet it forms the basis for one of two paragraphs of the discussion. The authors also performed a "systemic review" to find 4 papers that are relevant to serum folate levels and capecitabine toxicity. It is not clear why a systemic review is needed. Discussion of relevant work, including those papers, in the discussion would have been more appropriate. Treating a literature search for supporting information as a systemic review instead of using it as a means to place data in the context of the field through discussion is a missed opportunity and detracts from the case report further. The following are specific notes regarding major issues noted while reviewing the manuscript:

Extensive language editing is needed.

The title should be revised. As written, it is unclear and reads as though capecitabine worsens adverse events associated with capecitabine, which does not make sense. Also, two cases of toxicity are presented, not a series. The systematic review aspect also does not appear to meet the criteria of a systematic review, even if a PRISMA flow diagram is included.

The abstract should be revised to conform to the publisher's standards. While a structure abstract is not needed, the abstract lacks any structure or logical flow. The information that would normally be found in an abstract (background, study goals, results, conclusions) are absent.

It is generally preferable to not refer to people as being "wild type." The can be non-carriers for a given variant, but the definition of "wild type" for a human is unknown.

It is stated in the abstract that pharmacogenetic testing is routinely performed? where? for which drugs? for what genes/variants?

The authors choice of using online interaction references, instead of curated and peer reviewed sources is questioned. Why were those five databases chosen while others, which have the backing of being peer reviewed, were not?

What are capecitabine "SmPC reports?" Please define/explain.

The EMA guidance as far as pre-treatment testing for DPD deficiency is not as described by the authros. The authors are encouraged to make sure they adequately capture what the EMA document actually states.

The introduction could benefit from using a proper paragraph structure for flow. As written, it is difficult to follow the logic flow of the authors.

The rationale for the study is inadequately developed in the introduction.

The maximum dose of capecitabine is stated, but it is unclear how that relates to the information presented elsewhere. Did patients receive more than the maximum dose? Based on the case presentations, it does not appear so. Why is this here?

A list of genes containing variants associated with fluoropyrimide toxicity to varying degrees is presented in the intro. It is unclear as written how this information could be used, and it does not appear that it fully explains the observed toxicities.

In the first sentence of the results, what is meant by "pharmacological analysis?"

The reader will not know what SNPs are "routinely analyzed at the clinical pharmacology unit of the university hospital."

Two cases are presented. In the first, the patient experienced grade 3 diarrhea and HFS, which improved following a reduction in capecitabine and oxaliplatin dose (standard clinical response to drug-related toxicity). In the second case, the patient developed grade 4 HFS after 15 cycles of treatment with capecitabine, prompting discontinuation of capecitabine. Concomitant chemotherapy is not clear for this case (assuming monotherapy), but other medications and supplements are listed.

The clinical presentation of case 2 is confusing and indicates that the patient might have been mis-treated due to incorrect tumor staging. The patient is listed as having stage II colon cancer in the text and table, but metastases were detected by CT, which would indicate that the patient was mis-staged and instead had stage IV metastatic disease. If the detection of metastases occurred later during or after therapy, which would be considered advancement of disease, that is certainly not clear at all in the text.

What are the "recommended DPYD SNPs." Based on who's recommendation? One of the SNPs listed in the table has not been linked to 5-FU toxicity.

It is indicated earlier in the manuscript that patients were genotyped using some sort of panel that is in place at this hospital, yet table 2 shows that the two patients were not genotyped for the same panel of SNPs, please reconcile this discrepancy. The methods add further confusion because it seems like standard genotyping panels were not used. If not, what controls were in place?

The "causality assessment" is not adequately described in the materials and methods, and it is not clear what is actually being considered or evaluated. This type of method usually detects if a change in symptoms is temporally related to a change in treatment. Since capecitabine is already linked to the toxicities that were experienced, it is not clear what is being added by doing that type of analysis in this case.

There are formatting errors with the table and text on page 7.

The relevance of lipidic profiles is unclear. Please explain in the results why the data are being shown.

Folate is known to enhance toxicity in 5-FU treatment and forms of folate are typically co-administered with the drug as part of a multi-drug regimen (e.g., FOLFOX) and links between high serum folate levels and increased 5-FU/capecitabine ADRs have been reported previously. In fact, the use of folates is specifically not recommended with capecitabine therapies because of the known high risk of toxicity. This is basic clinical knowledge that should be known without consulting online resources like WebMD.

It is unclear what is added by including pictures of HFS.

The authors performed a literature review effectively searching for studies that evaluated folate and capecitabine. They identified 4. Why is this presented as a systemic review, as opposed to just discussing the relevant literature in the discussion. It is not clear what the systemic review aspect adds to the paper because the relevance of the identified papers is not adequately developed in the discussion.

Please define "DDI."

Gene abbreviations should be italicized consistent with HGNC naming throughout.

The discussion could also benefit from having some structure. The first paragraph spans lines 301 to 388 across nearly 2 full pages. It jumps from topic to topic without any transition or reference points.

The discussion of MTHFR genotypes is confusing as well, since only 1 of the patients seems to carry a SNP in MTHFR. What are the references for the importance of these SNPs? How strong is the evidence that they are linked to toxicity? Later it is stated that more data are needed regarding the links between MTHFR and toxicity (without any sort of discussion of the underlying data/studies), which overstates their importance as biomarkers.

The discussion of TS/TYMS/TSER variants is also out of place since both patients are 2R/2R. Why is 2R/3R being discussed as though they could potentially contribute to the toxicity observed? Where is the citation for the statistic claiming a 20x increased risk for toxicity in 2R/2R vs 3R/3R. With the standard toxicity rate in 5-FU/capecitabine treatment being ~20-35%, this doesn't seem possible to have an RR of 20. Regardless, the 2R vs 3R data are not nearly as strong in the literature as the authors are suggesting, and the "2R"-style nomenclature misses important details about the TSER variants. Specifically, it is not the repeat structure that defines (weak) toxicity associations, but instead the number of transcription factor binding sites, which is determined by a combination of repeat number and SNP status within each repeat.

The statement that the TSER 2R genotype is more prevalent in female cancer patients is also very confusing. The authors of the cited paper indicate that sex might contribute to TSER associations with toxicity, but concluding that there is a sex-specific difference in prevalence in cancer patients is a mis-interpretation of the literature.

The conclusions drawn from the genetic and non-genetic data in these two cases are pure speculation and without support (discussion section).

The statements leading up to the one about the WHO and that statement are unclear. What are the authors trying to convey? It is indicated in the subsequent sentences that medication discrepancies during transition between care facilities may have occurred. Was this the case here, and why is it first being mentioned this late in the discussion instead of during the case presentation?

The conclusion, as stated by the authors in the conclusions section, is unclear and does not seem to rationally flow from the data presented for the two cases. The two cases of toxicity can be readily explained by knowns in the field and by information that is effectively on the drug label.